# Improved lipidomic profile mediates the effects of adherence to healthy lifestyles on coronary heart disease

Jiahui Si[1,2], Jiachen Li[1], Canqing Yu[1,3], Yu Guo[4], Zheng Bian[4], Iona Millwood[5,6], Ling Yang[5,6], Robin Walters[5,6], Yiping Chen[5,6], Huaidong Du[5,6], Li Yin[7], Jianwei Chen[8], Junshi Chen[9], Zhengming Chen[6], Liming Li[1,3]*, Liming Liang[2]*, Jun Lv[1,3,10]*

[1]Department of Epidemiology and Biostatistics, School of Public Health, Peking University Health Science Center, Beijing, China; [2]Departments of Epidemiology and Biostatistics, Harvard T.H. Chan School of Public Health, Boston, United States; [3]Peking University Institute of Public Health & Emergency Preparedness, Beijing, China; [4]Chinese Academy of Medical Sciences, Beijing, China; [5]Medical Research Council Population Health Research Unit at the University of Oxford, Oxford, United Kingdom; [6]Clinical Trial Service Unit & Epidemiological Studies Unit (CTSU), Nuffield Department of Population Health, University of Oxford, Oxford, United Kingdom; [7]NCDs Prevention and Control Department, Hunan Center for Disease Control & Prevention, Changsha, China; [8]Liuyang Center for Disease Control & Prevention, Liuyang, Hunan, China; [9]China National Center for Food Safety Risk Assessment, Beijing, China; [10]Key Laboratory of Molecular Cardiovascular Sciences (Peking University), Ministry of Education, Beijing, China

*For correspondence:
lmleeph@vip.163.com (LL);
lliang@hsph.harvard.edu (LL);
epi.lvjun@vip.163.com (JL)

Competing interests: The authors declare that no competing interests exist.

**Abstract** Adherence to healthy lifestyles is associated with reduced risk of coronary heart disease (CHD), but uncertainty persists about the underlying lipid pathway. In a case–control study of 4681 participants nested in the prospective China Kadoorie Biobank, 61 lipidomic markers in baseline plasma were measured by targeted nuclear magnetic resonance spectroscopy. Baseline lifestyles included smoking, alcohol consumption, dietary habit, physical activity, and adiposity levels. Genetic instrument was used to mimic the lipid-lowering effect of statins. We found that 35 lipid metabolites showed statistically significant mediation effects in the pathway from healthy lifestyles to CHD reduction, including very low-density lipoprotein (VLDL) particles and their cholesterol, large-sized high-density lipoprotein (HDL) particle and its cholesterol, and triglyceride in almost all lipoprotein subfractions. The statins genetic score was associated with reduced intermediate- and low-density lipoprotein, but weak or no association with VLDL and HDL. Lifestyle interventions and statins may improve different components of the lipid profile.

## Introduction

Coronary heart disease (CHD) has become one of the leading causes of death worldwide (*Roth et al., 2018*). In China, the mortality rate from CHD increased almost four times from 2002 to 2014 (*Chen et al., 2017*). It is widely acknowledged that unhealthy lifestyles, such as smoking, excess alcohol consumption, inadequate physical activity, unhealthy diet, and adiposity, are major risk factors for CHD (*Dariush et al., 2008*). Studies on the impacts of adherence to a combination of healthy lifestyle factors (HLFs) on mortality (*Li et al., 2018*; *Zhu et al., 2019*), healthy life expectancy (*Li et al., 2020*), and risk of type 2 diabetes (*Lv et al., 2017a*) and cardiovascular diseases in the

Chinese population (*Lv et al., 2017b*) have provided important information on the maximum public health benefit that lifestyle intervention could achieve.

Atherogenic dyslipidemia is one of the well-documented risk factors for CHD (*Shaima et al., 2016*; *Peters et al., 2016*). Conventional lipid markers fail to distinguish between the size, density, concentration, or composition of lipoprotein particles, which may have contrasting effects on CHD risk (*Holmes et al., 2018*; *Würtz et al., 2015*). Lipidomics provides a detailed snapshot of the systemic lipid profile beyond routine lipid markers. Only a few studies have examined the association between lipidomic profile and individual HLFs separately (*Kujala et al., 2013*; *Würtz et al., 2016a*; *Würtz et al., 2014*; *Lankinen et al., 2014*), which, however, typically correlates with one another. It is mostly unknown how much of the effects of combined HLFs on reduced CHD risk are mediated through an improved lipid profile and what the differences are between components of the lipid profile in their mediating effects.

Statins are HMG-CoA reductase (HMGCR) inhibitors, which reduce the low-density lipoprotein cholesterol (LDL-C) by interfering with the cholesterol-biosynthetic pathway and have become one of the first-line therapy options for dyslipidemia (*Sirtori, 2014*). Mendelian randomization studies constructed in the European population observed lipid-lowering effect of statins beyond the anticipated decrease in LDL particles (*Ference et al., 2019*; *Würtz et al., 2016b*). However, such genetic effects have not been examined in Asian populations. No study compares the effects of healthy lifestyle and genetically inferred lipid-lowering medications on lipidomic profile in the same set of study participants.

The primary aims of the present study were to examine the combined effect of HLFs on components of a comprehensive lipidomic profile measured by nuclear magnetic resonance (NMR) spectroscopy (*Soininen et al., 2015*), and further quantify how much of the combined effects of HLFs on CHD reduction are mediated through lipid metabolites. We also estimated the clinical effect of statins and bempedoic acid, a novel therapeutic approach by inhibiting ATP citrate lyase (ACLY) (*Pinkosky et al., 2016*) on lipidomic profile by creating a Chinese specific genetic score for HMGCR and ACLY functions. Finally, we examined the joint effects of HLFs and lipid-lowering medications on lipidomic profile. We did a nested case–control study comprising incident CHD cases, stroke cases, and controls identified from the 10-year follow-up of the China Kadoorie Biobank (CKB). We included all eligible participants to examine the impact of HLFs and genetic scores on lipid metabolites, and only included CHD cases and controls in the further mediation analysis.

## Results

The mean age of 4681 participants was 46.7 ± 8.0 years. Five HLFs included never smoking, moderate alcohol consumption, having a healthy dietary score ≥4, being physically active, and healthy adiposity levels. Of the 4681 participants, 0.2%, 11.1%, and 47.4% had at least 5, 4, and 3 HLFs, respectively. The overall mean (SD) LDL-C concentration measured by the clinical chemistry assay was 88.8 (27.0) mg/dl. Younger, female, and more educated participants were more likely to adopt a healthy lifestyle (*Table 1*). Compared with control participants, the CHD cases were older, were less likely to be women, and had a higher prevalence of hypertension and diabetes at baseline (*Supplementary file 2A*).

### Associations of combined HLFs with lipid metabolites

Adherence to combined HLFs was associated with 50 components of the lipid profile (false dicovery rate [FDR] < 0.05). Compared with participants who adopted at most one HLF, the differences in the lipid metabolites, especially VLDL- and HDL-related measures, increased with the number of HLFs adhered to (*Figures 1–3*). Participants with four to five HLFs had lower VLDL particle concentrations and smaller VLDL particle size, with adjusted SD difference (95% CI) ranging from −0.54 (−0.66,−0.43) for large VLDL to −0.27 (−0.38,−0.15) for very small VLDL, and −0.47 (−0.59,−0.36) for VLDL diameter (*Figure 1*, *Figure 1—source data 1*). For HDL, adherence to four to five HLFs was associated with higher HDL particle concentrations and larger HDL particle size, with maximum SD difference (95% CI) of 0.39 (0.28, 0.50) for large HDL, and 0.32 (0.21, 0.43) for HDL diameter. The corresponding SD difference (95% CI) for apolipoprotein B/apolipoprotein A1 was −0.45 (−0.56,−0.34), resulting from higher apolipoprotein A1 and lower apolipoprotein B.

**Table 1.** Age-, sex-, and study area-adjusted baseline characteristics of 4681 participants according to the number of healthy lifestyle factors (HLFs).

The results are presented as adjusted means or percentages, with adjustment for age, sex, and study area, as appropriate. All baseline characteristics were associated with the number of HLFs, with p<0.05 for trend across categories, except for urban or rural residence (0.155), family history of heart attack (p=0.905), and consumption of fresh vegetables (0.065).

| Baseline characteristics | 0 | 1 | 2 | 3 | $\geq$4 |
|---|---|---|---|---|---|
| No. of participants, n (%) | 118 (2.5) | 688 (14.7) | 1656 (35.4) | 1698 (36.3) | 521 (11.1) |
| Age, year | 49.6 | 49.0 | 47.5 | 45.8 | 43.8 |
| Female, % | 5.1 | 22.6 | 46.4 | 62.9 | 67.5 |
| Urban area, % | 42.6 | 33.5 | 27.8 | 25.1 | 36.6 |
| Middle school and above, % | 51.5 | 53.6 | 53.8 | 57.9 | 58.2 |
| Married, % | 91.3 | 92.9 | 94.6 | 95.3 | 95.2 |
| Prevalent hypertension, % | 62.6 | 52.6 | 48.7 | 40.6 | 38.5 |
| Prevalent diabetes, % | 13.8 | 12.0 | 7.1 | 4.1 | 3.8 |
| Family history of heart attack, % | 2.8 | 5.0 | 4.7 | 4.2 | 4.8 |
| Having HLFs*, % | | | | | |
| Never smoking | – | 47.1 | 57.0 | 70.6 | 85.9 |
| Moderate alcohol consumption | – | 3.5 | 8.6 | 15.9 | 28.3 |
| Being physically active | – | 13.3 | 37.6 | 66.2 | 96.0 |
| Healthy dietary pattern | – | 23.9 | 41.4 | 60.2 | 94.0 |
| Vegetables 7 days/week | 90.8 | 93.2 | 92.5 | 93.0 | 97.6 |
| Fruit 7 days/week | 2.6 | 7.6 | 10.2 | 16.3 | 24.8 |
| Read meat <7 days/week | 52.9 | 65.7 | 73.7 | 76.7 | 84.8 |
| Soybean product $\geq$4 days/week | 2.4 | 4.0 | 7.0 | 10.7 | 19.2 |
| Fish $\geq$1 day/week | 18.3 | 23.9 | 30.3 | 38.5 | 50.3 |
| Coarse grains $\geq$4 days/week | 7.9 | 21.1 | 22.7 | 23.7 | 25.0 |
| Healthy adiposity level | – | 25.8 | 55.5 | 89.0 | 98.0 |

*HLFs were defined as: never smoking; weekly but not daily drinking or daily drinking less than 30 g of pure alcohol; engaging in a sex-specific median or higher level of physical activity; engaging in more than or equal to 4 of total six healthy diet components; having a body mass index between 18.5 and 27.9 kg/m$^2$ and having a waist circumference <90 cm in men and <85 cm in women.

The associations of combined HLFs with cholesterol concentrations in lipoprotein subfractions were very similar to the associations with the corresponding lipoprotein particle concentrations (*Figure 2*, *Figure 2—source data 1*). Combined HLFs were consistently associated with lower TG concentrations in all lipoprotein subfractions except large HDL particles. The adjusted SD difference (95% CI) for participants with four to five HLFs ranged from −0.55 (−0.66,–0.43) for small VLDL-TG to −0.29 (−0.41,–0.18) for medium LDL-TG (*Figure 3*, *Figure 3—source data 1*).

The linear associations between each one factor increase in HLFs and lipid profile were illustrated in *Figure 1—figure supplement 1*. In sensitivity analyses, we further adjusted for prevalent diabetes, restricted analyses to control participants (*Supplementary file 2B*), did not adjust for fasting time, used more strict body mass index (BMI) and waist circumference (WC) cut-off points to define healthy adiposity, or excluded moderate alcohol consumption from the HLF definition; the associations between HLFs and metabolites were not substantially altered (*Figure 1—figure supplements 2–4*).

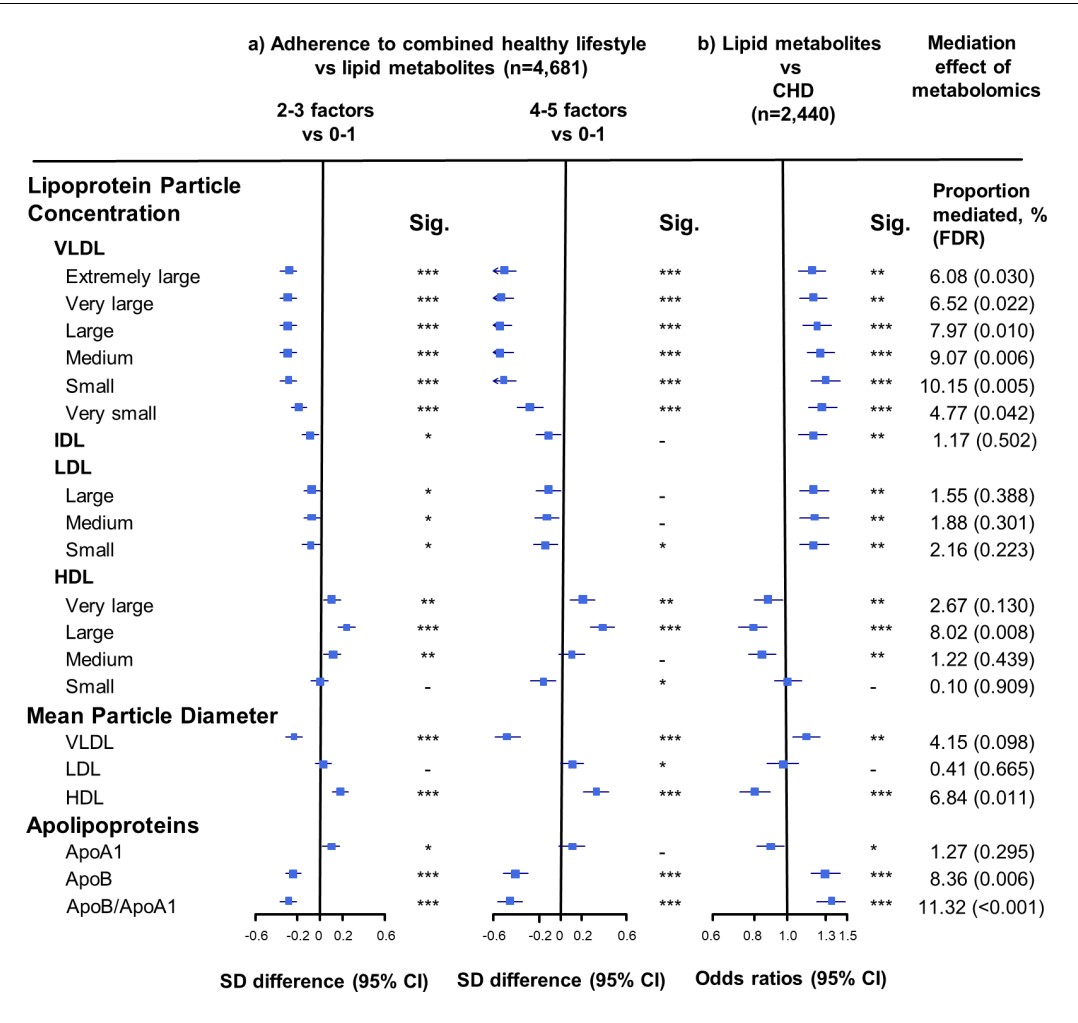

**Figure 1.** Associations of size-specific lipoprotein particle concentrations, mean lipoprotein particle diameter, and apolipoprotein concentrations with combined healthy lifestyle and risk of coronary heart disease. (a) SD difference and 95% CI are for comparison of participants who adopted two to three or four to five combined healthy lifestyles with participants who adopted zero to one. Multivariable model was adjusted for: age, sex, fasting time, study areas, education level, and case/control status. (b) Odds ratio and 95% CI are for the associations of 1-SD metabolic markers increasing with CHD risk. Multivariable model was adjusted for: age, sex, fasting time, study areas, education level, and smoking status. Horizontal lines represent 95% CIs. ApoA1 = apolipoprotein A1; ApoB = apolipoprotein B; CHD = coronary heart disease; HDL = high-density lipoprotein; IDL = intermediate-density lipoprotein; LDL = low-density lipoprotein; Sig. = significance ***p≤0.0001, **p≤0.01, *p≤0.05, – p>0.05 (false discovery rate [FDR]–adjusted p-values); VLDL = very low-density lipoprotein.

The online version of this article includes the following source data and figure supplement(s) for figure 1:

**Source data 1.** Associations of size-specific lipoprotein particle concentrations, mean lipoprotein particle diameter, and apolipoprotein concentrations with combined healthy lifestyle and risk of coronary heart disease.

**Figure supplement 1.** Associations between number of healthy lifestyles and lipid metabolites.

**Figure supplement 2.** Associations between number of healthy lifestyles and lipid metabolites without adjustment for fasting time.

**Figure supplement 3.** Associations between number of healthy lifestyles and lipid metabolites with a more strict definition of healthy adiposity (body mass index in the range of 18.5–24.9 kg/m² and waist circumference <90 cm in men and <80 cm in women).

**Figure supplement 4.** Associations between number of healthy lifestyles and lipid metabolites without including moderate alcohol consumption as a healthy lifestyle.

**Figure supplement 5.** Associations between moderate alcohol consumption and lipid metabolites.

**Figure supplement 6.** Associations between being physically active and lipid metabolites.

*Figure 1 continued on next page*

*Figure 1 continued*

**Figure supplement 7.** Associations between healthy adiposity level (body mass index in the range of 18.5–27.9 kg/m$^2$ and waist circumference <90 cm in men and <85 cm in women) and lipid metabolites.

**Figure supplement 8.** Associations between healthy adiposity level (body mass index in the range of 18.5–24.9 kg/m$^2$ and waist circumference <90 cm in men and <80 cm in women) and lipid metabolites.

**Figure supplement 9.** Associations of alcohol consumption with size-specific lipoprotein particle concentrations, mean lipoprotein particle diameter, and apolipoprotein concentrations.

**Figure supplement 10.** Associations of alcohol consumption with cholesterol concentrations in lipoprotein subfractions.

**Figure supplement 11.** Associations of alcohol consumption with triglyceride concentrations in lipoprotein subfractions.

**Figure supplement 12.** Associations between never smoking and lipid metabolites.

**Figure supplement 13.** Associations between healthy diet pattern and lipid metabolites.

## Associations of individual HLFs with lipid metabolites

Of the five individual HLFs analyzed, moderate alcohol consumption (*Figure 1—figure supplement 5*), being physically active (*Figure 1—figure supplement 6*), and having healthy adiposity levels (*Figure 1—figure supplement 7*), had the most significant influence on lipid metabolites. Participants who were physically active or had healthy adiposity levels had a cardioprotective lipid profile, with lower concentrations of VLDL-related measures, apolipoprotein B, and higher concentrations of larger HDL particles. The maximum SD differences (95% CI) related to physical activity (*Figure 1—figure supplement 6*) and healthy adiposity level (*Figure 1—figure supplement 7*) were −0.12 (−0.18,–0.06) for medium VLDL-TG and −0.54 (−0.60,–0.48) for total TG, respectively. Sensitivity analysis using more strict BMI and WC cut-off points (BMI in the range of 18.5–24.9 kg/m$^2$ and WC <90 cm in men and <80 cm in women) observed similar and generally stronger associations between healthy adiposity level and lipidomic profile (*Figure 1—figure supplement 8*).

Moderate alcohol consumption was associated with higher concentrations of VLDL- and HDL-related measures and apolipoprotein A1. The maximum SD difference (95% CI) was 0.29 (0.19, 0.38) for small HDL particle concentration (*Figure 1—figure supplement 5*). We further divided participants into three groups according to their alcohol consumption at baseline: non-regular, moderate, and heavy use. Compared with non-regular use group, both heavy (with ≥30 g of pure alcohol per day) and moderate alcohol use (<30 g per day) had a similar pattern of effects on lipid metabolites, with the most significant changes observed in participants with heavy alcohol use (*Figure 1—figure supplements 9–11*).

Smoking (*Figure 1—figure supplement 12*) and dietary habit (*Figure 1—figure supplement 13*) had a relatively small impact on lipid metabolites.

## Mediation effects of lipid metabolites in the association between HLFs and CHD risk

We restricted the following analyses in 927 incident CHD cases and 1513 controls. Incident CHD cases were those who developed fatal ischemic heart disease and nonfatal myocardial infarction during follow-up. The associations between lipid metabolites and CHD risk generally mirrored the associations between combined HLFs and lipid metabolites (*Figures 1–3*). None of the lipid metabolites showed interactions with the HLFs in their effect on CHD risk (all $p_{interation}$ > 0.05). A total of 35 lipid metabolites showed statistically significant mediation effects from combined HLFs to CHD reduction (FDR ranging from <0.001 to 0.042). The proportions of reduced CHD risk associated with combined HLFs mediated by VLDL particle concentration ranged from 4.77% for very small VLDL to 10.15% for small VLDL (*Figure 1*, *Figure 1—source data 1*). Other strong mediators included large HDL (8.02%), apolipoprotein B (8.36%), and apolipoprotein B/apolipoprotein A1 (11.32%). For cholesterol, compared to LDL-C, VLDL- and HDL-C were relatively strong mediators (*Figure 2*, *Figure 2—source data 1*). TG carried within all lipoproteins (except for large-sized HDL) showed statistically significant mediating effects, with the maximum proportion of 10.47% for small VLDL-TG (*Figure 3*, *Figure 3—source data 1*). The top five principal components of all lipid metabolites mediated 14.05% of the reduced CHD risk associated with combined HLFs.

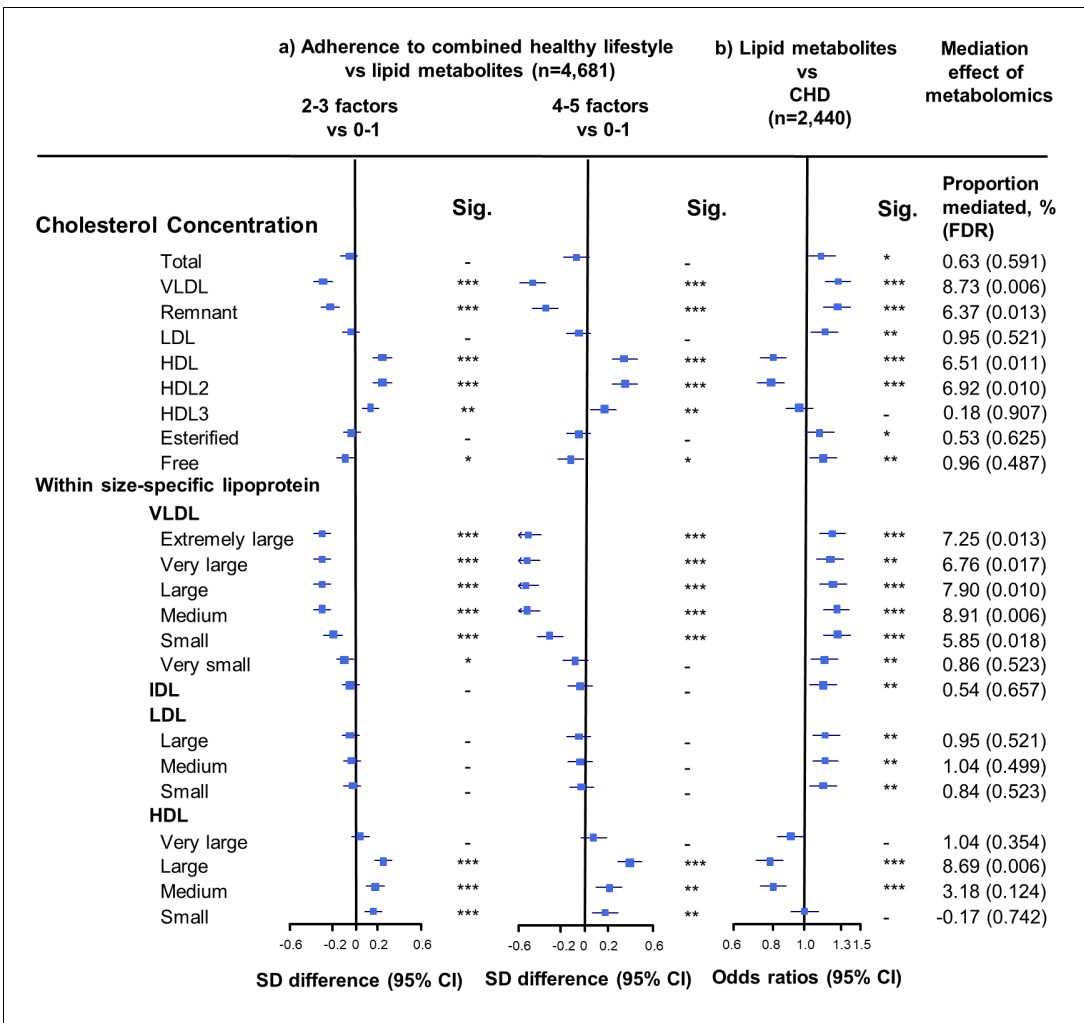

**Figure 2.** Associations of cholesterol concentrations in lipoprotein subfractions with combined healthy lifestyle and risk of coronary heart disease. (a) SD difference and 95% CI are for comparison of participants who adopted two to three or four to five combined healthy lifestyles with participants who adopted zero to one. Multivariable model was adjusted for: age, sex, fasting time, study areas, education level, and case/control status. (b) Odds ratio and 95% CI are for the associations of 1-SD metabolic markers increasing with CHD risk. Multivariable model was adjusted for: age, sex, fasting time, study areas, education level, and smoking status. Horizontal lines represent 95% CIs. CHD = coronary heart disease; HDL2 = larger HDL particles; HDL3 = smaller HDL particles; Sig. = significance ***p≤0.0001, **p≤0.01, *p≤0.05, – p>0.05 (false discovery rate [FDR]–adjusted p-values); other abbreviations as in *Figure 1*.

The online version of this article includes the following source data for figure 2:

**Source data 1.** Associations of cholesterol concentrations in lipoprotein subfractions with combined healthy lifestyle and risk of coronary heart disease.

## HMGCR and ACLY scores, HLFs, and lipid metabolites

The *HMGCR* and *ACLY* scores had a similar pattern of effects on lipid metabolites, with higher scores mainly associated with decreased concentrations of intermediate-density lipoprotein (IDL)- and LDL-related measures and apolipoprotein B (*Supplementary file 2C and D*). The sum of *HMGCR* and *ACLY* scores was associated with stronger changes in the above lipid metabolites (*Supplementary file 2E*). Use of genetic scores based on the European population (*Ference et al., 2019*) for *HMGCR* and *ACLY* observed similar but weaker associations (*Supplementary file 2C–F*).

In the joint association analysis of HLFs and *HMGCR* score with lipid metabolites, compared with participants who had higher genetic risk (median cutoffs) and adhered to zero to two HLFs, those with lower genetic risk and three to five HLFs had the most cardioprotective lipidomic profile,

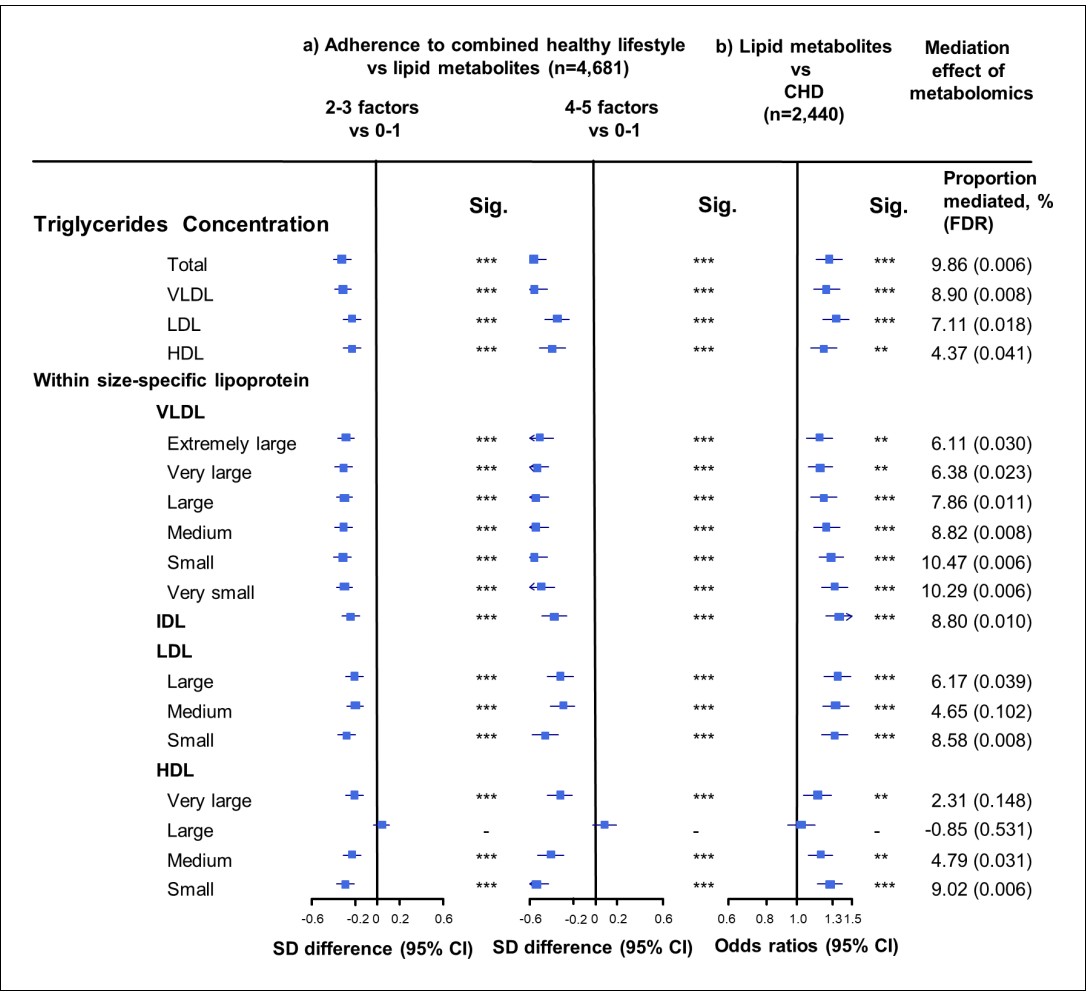

**Figure 3.** Associations of triglyceride concentrations in lipoprotein subfractions with combined healthy lifestyle and risk of coronary heart disease. (a) SD difference and 95% CI are for comparison of participants who adopted two to three or four to five combined healthy lifestyles with participants who adopted zero to one. Multivariable model was adjusted for: age, sex, fasting time, study areas, education level, and case/control status. (b) Odds ratio and 95% CI are for the associations of 1-SD metabolic markers increasing with CHD risk. Multivariable model was adjusted for: age, sex, fasting time, study areas, education level, and smoking status. Horizontal lines represent 95% CIs. CHD = coronary heart disease; Sig. = significance ***p≤0.0001, **p≤0.01, *p≤0.05, – p>0.05 (false discovery rate [FDR]–adjusted p-values); Abbreviations as in *Figure 1*.

The online version of this article includes the following source data for figure 3:

**Source data 1.** Associations of triglyceride concentrations in lipoprotein subfractions with combined healthy lifestyle and risk of coronary heart disease.

including 0.36 SD decrease in VLDL-C, 0.13 SD decrease in LDL-C, and 0.21 SD increase in HDL-C (*Figure 4* and *Figure 4—figure supplement 1* for *ACLY* score). We further compared the effect patterns of each one factor increase in HLFs with a 2-SD increase in *HMGCR* or *ACLY* score on lipid metabolites (*Figure 4—figure supplements 2* and *3*). The combined HLFs, as opposed to the effect by *HMGCR* and *ACLY* scores, were associated with lower VLDL-related measures, apolipoprotein B, and TG in almost all lipoprotein subfractions, and with higher HDL and HDL-C concentrations.

When we stratified participants according to the score of *HMGCR*, *ACLY*, or their sum score, the associations between each one factor increase in HLFs and lipid metabolites were generally similar between high- and low- genetic risk stratum (all $p_{interaction}$ >0.05) (*Supplementary file 2C–E*).

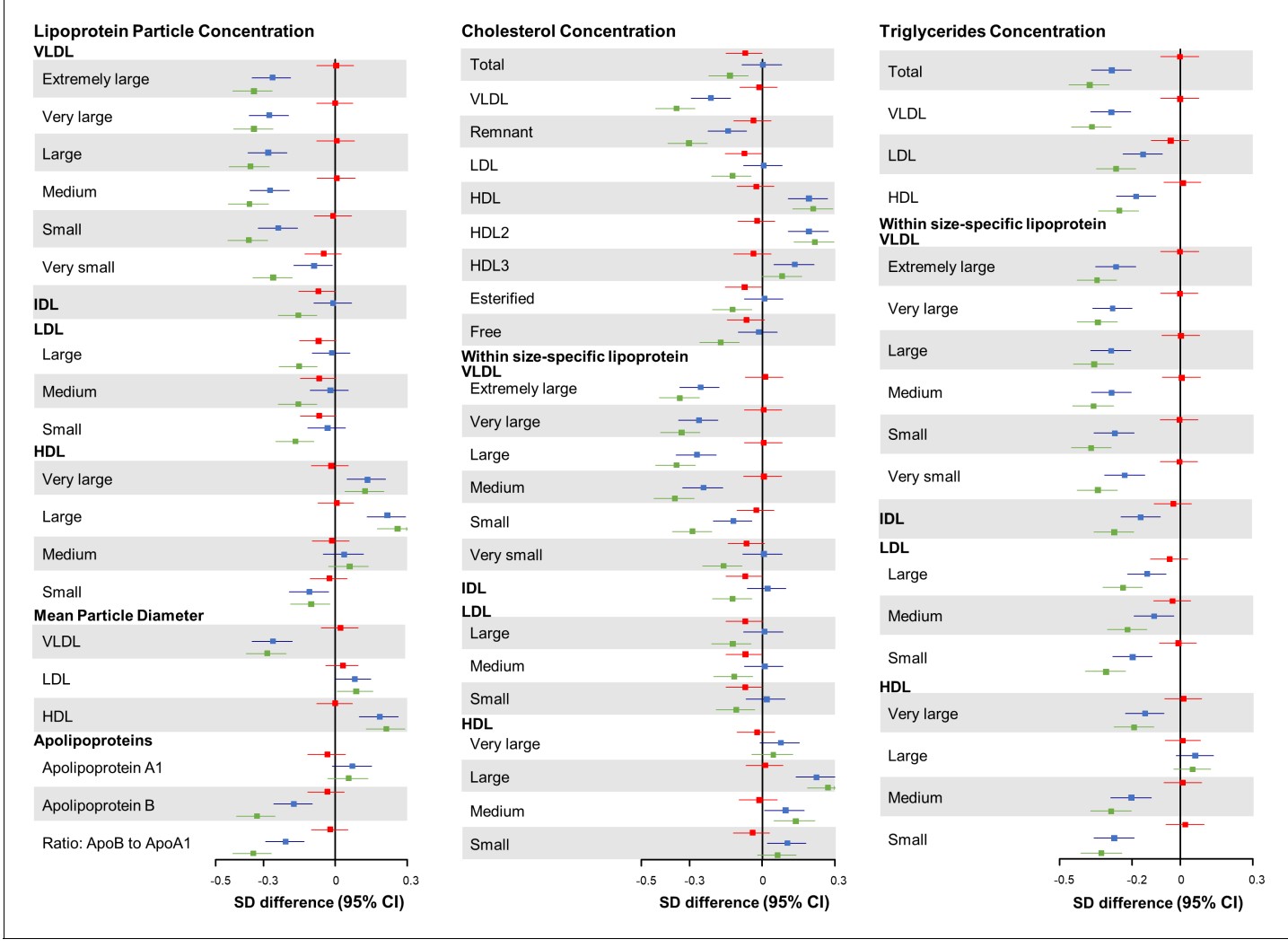

**Figure 4.** Joint association of combined healthy lifestyle and *HMGCR* scores based on Chinese population with lipid metabolites. Participants who had higher genetic risk regarding HMGCR (3-hydroxy-3-methylglutaryl–coenzyme A reductase) and adhered to zero to two healthy lifestyle factors (HLFs) were reference group. SD difference and 95% CI of log-transformed lipid metabolites for participants with lower genetic risk and 0-2 HLFs, higher genetic risk and 3-5 HLFs, and lower genetic risk and 3-5 HLFs were shown in red, blue, and green, respectively. Abbreviations as in *Figure 1*. The online version of this article includes the following figure supplement(s) for figure 4:

**Figure supplement 1.** Joint association of combined healthy lifestyle and *ACLY* score based on Chinese population with lipid metabolites.

**Figure supplement 2.** Associations of combined healthy lifestyle and *HMGCR* score based on Chinese population with changes in the lipid metabolites.

**Figure supplement 3.** Associations of combined healthy lifestyle and ACLY score based on Chinese population with changes in the lipid metabolites.

## Discussion

In this prospective study of middle-aged Chinese, participants who adhered to healthy lifestyles tended to have a more cardioprotective lipidomic profile, which jointly mediated 14% of the protective effect of combined HLFs on CHD reduction. Similar to results in the European population, genetic scores for the targets of statins and ACLY inhibitors showed similar effects on reducing concentrations of IDL- and LDL-related measures, while the underlying mechanisms of lifestyle intervention were more strongly related to VLDL- and HDL-related measures, apolipoprotein B, and TG in almost all lipoprotein subfractions.

Our findings on the associations of lipid metabolites with individual lifestyle-related characteristics like physical activity, adiposity, and alcohol consumption were generally consistent with previous

studies (*Kujala et al., 2013*; *Würtz et al., 2016a*; *Würtz et al., 2014*). One of the studies used the Mendelian randomization to indicate causal adverse effects of increased adiposity on lipoprotein subclass profiles within the non-obese weight range among young Finland adults (*Würtz et al., 2014*). Another study similarly showed a mixture of favorable and adverse effects of alcohol consumption on the lipid profile in relation to cardiovascular disease (*Würtz et al., 2016a*). Also, the lipoprotein lipid profile observed cross-sectionally was highly consistent with the pattern of their changes accompanying a change in alcohol consumption at 6-year follow-up. Numerous studies have found increased HDL level with higher alcohol consumption (*Gepner et al., 2015*; *Brien et al., 2011*). However, the association between alcohol consumption and apolipoprotein B-carrying lipoprotein is less clear (*Brien et al., 2011*; *Roerecke and Rehm, 2014*). Our results noted that alcohol consumption showed divergent relationships with different sized apolipoprotein B-carrying particles, for example, higher concentration of large-sized VLDL and lower concentration of IDL and LDL. Although the explanation for the complex association between alcohol consumption and lipid profile remains inconclusive, our detailed investigation of lipoprotein subclasses provides improved understanding of the diverse molecular process related to alcohol consumption. Regarding diet, a previous randomized trial found that diet in rich of whole grain, bilberries, and fatty fish caused changes in HDL particles (*Lankinen et al., 2014*). The present study characterized a healthy diet differently and found significant differences in VLDL-related measures but not HDL.

This is the first study, to our knowledge, to assess the combined effect of HLFs on a comprehensive lipidomic profile. The results indicated that participants with healthy lifestyles were characterized by an antiatherogenic lipidomic profile, which has been related to lower CHD risk previously (*Varbo et al., 2013*; *Holmes et al., 2015*; *Peter et al., 2015*; *Natarajan et al., 2010*; *Inouye et al., 2010*). The positive associations of combined HLFs with HDL and HDL-C were limited to large and medium subclasses, in line with previous studies which suggested that small HDL particles did not have protective effects on CHD (*Peter et al., 2015*; *Natarajan et al., 2010*; *Inouye et al., 2010*). Notably, the TG levels within all apolipoprotein B and most HDL particles were lower in participants who adopted healthy lifestyles. It is plausible that healthy lifestyles have opposing relationships with HDL-TG and HDL-C.

Limited prospective studies have investigated the mediating effects of individual HLFs on CHD through total cholesterol, suggesting that it mediates 13%, 8%, and 18% of the excess CHD risk related to inadequate physical activity (*Mora et al., 2007*), obesity, and overweight (*Global Burden of Metabolic Risk Factors for Chronic Diseases Collaboration (BMI Mediated Effects), 2014*), respectively. Our results showed all NMR-measured lipid metabolites jointly explained 14% of the protective effect of combined HLFs on CHD risk. We further highlighted the differences between various lipid metabolites in their mediating effects. The apolipoprotein B/A1 ratio was among the most influential mediators and has been previously reported to be a better predictor of CHD risk than any of the cholesterol ratios (*McQueen et al., 2008*). A lower apolipoprotein B/A1 ratio, together with manifestation of other metabolites, suggested that adherence to HLFs can reduce the risk of CHD through both lower proatherogenic and higher antiatherogenic lipoproteins.

Both statins and ACLY inhibitors have been associated with lowering LDL- and IDL-related measures and further with a reduction in CVD risk in western populations (*Ference et al., 2019*; *Würtz et al., 2016b*). In the present study, we used genetic scores to mimic the effects of these two LDL-C lowering targets and observed similar effects on the lipidomic profile. For LDL and IDL particle concentrations, adherence to two or more HLFs could achieve a similar beneficial effect as a 2-SD change in *ACLY*/*HMGCR* genetic scores. In other words, two or more HLFs could compensate for the deleterious effect on lipid metabolites due to inheriting risk alleles in these genes. More importantly, we found that adherence to combined HLFs had a much stronger effect on other components of the lipidomic profile than LDL- and IDL-related measures, including VLDL- and HDL-related components.

To our knowledge, this is the first study to reveal the potential underlying lipid pathways that may mediate the effects of adherence to combined HLFs on lower CHD risk. The strengths of the study include the prospective outcome ascertainment, a comprehensive assessment of lifestyle factors, and a population free of lipid-modifying therapy at the time of blood collection. The measurements of multiple lipids and lipoprotein particles provide a detailed snapshot of the systemic lipid profile. The concordance of measurements by both NMR spectroscopy and clinical chemistry assays, and by duplicate samples of NMR metabolomics provided evidence to support internal validity. The

availability of genotyping data allowed us to use a Mendelian randomization approach to estimate the effects of statins and ACLY inhibition on lipid metabolites while avoiding potential confounding bias by indication.

Our study has limitations. First, the lifestyle behaviors were self-reported once at baseline. Second, lifestyle behaviors and lipid metabolites were measured at the same time. However, previous evidence supports the causative effects of individual HLFs on lipid metabolites and the resemblance between the cross-sectional and longitudinal association patterns (*Würtz et al., 2016a*; *Würtz et al., 2014*). Third, the lipid metabolites quantified by the NMR spectroscopy assay did not include some important measures such as lipoprotein(a), apolipoprotein CIII, and HDL functionality. There are also strong correlations between the lipid metabolites, with multiple measures representing the same underlying lipid fractions. As a result, the mediating role of lipid metabolites in the present study cannot extrapolate to that of the complete lipidomic profile and also cannot differentiate the individual mediating role of each lipid metabolites. Nevertheless, this does not detract from the value of the study in identifying potential pathways underlying the HLFs and CHD risk. Fourth, the effect of therapeutic agents mimicked by genetic scores is the effect of lifelong exposure to a biomarker on an outcome that is difficult to be translated into the expected effect of short-term pharmacologic changes (*Ference et al., 2019*). However, genetic scores served mainly for comparison of the underlying mechanisms of lifestyle interventions and lipid-lowering medications, rather than the effect size. Lastly, a more sophisticated HLFs score with appropriated weight might show stronger association. However, a more straightforward definition would be easier to understand and adapted by the public. Also, mediation analysis might be biased when the continuous exposure variable was dichotomized. Our simulation showed that this requires a particularly strong association between exposure and mediator, which was far from the realistic association between HLFs and metabolite in our study.

The present study of Chinese adults elucidated that the effects of adherence to a combination of HLFs on lower CHD risk were partly mediated by an improved lipid profile. Lifestyle interventions and lipid-lowering medication therapies may affect different components of the lipid profile, suggesting that they are not redundant strategies but could be combined for better benefits.

## Materials and methods

### Study population

The CKB is a prospective cohort of 512,715 adults (aged 30–79 years) from 10 geographically diverse areas across China (five urban sites and five rural sites) during 2004–2008. Details of the study design, survey methods, and long-term follow-up have been given elsewhere (*Chen et al., 2005*; *Chen et al., 2011*). Briefly, all participants had baseline data collected by questionnaire, including sociodemographic, lifestyle factors, and medical and medication history, and physical measurements. Participants also provided a 10 ml random blood sample for long-term storage, with the time since last meal recorded. Mortality and morbidity during follow-up were identified through linkage with local death and disease registries, with the national health insurance system, and by active follow-up if necessary (i.e., visiting local communities or directly contacting participants). Since 2014, 97% of the participants have been linked to the health insurance databases. By December 31, 2015, of all the cohort participants, only 4875 (<1%) were lost to follow-up. The mean follow-up duration of the cohort since baseline was 9.2 (1.4) years.

The study protocol was approved by the Ethics Review Committee of the Chinese Center for Disease Control and Prevention (005/2004, Beijing, China) and the Oxford Tropical Research Ethics Committee, University of Oxford (025–04, UK). All participants provided written informed consent.

### Design of the present study

A subset of 4681 CKB participants was selected for metabolomics measurements in a nested case–control study of incident CHD and stroke occurring before the censoring date of January 1, 2015 (*Holmes et al., 2018*). Cases were those who had a newly developed fatal or nonfatal disease during follow-up: (1) CHD: fatal ischemic heart disease coded as ICD-10 I20-I25 and nonfatal myocardial infarction coded as I21-I23 (n = 927); (2) ischemic stroke: ICD-10 I63 or I69.3 (n = 1114); (3) intracerebral hemorrhage: ICD-10 I61 or I69.1 (n = 1127). Case status was defined as the disease first

occurred in each participant. Common controls were selected by frequency matching to combined cases by age, sex, and study area (n = 1513). The diagnosis adjudication has finished for 34,000 reported cases of ischemic heart disease by a review of hospital medical records. Overall, 88% of the diagnoses were confirmed. All case and control participants did not report doctor-diagnosed CHD, stroke, transient ischemic attack, or cancer, and were not using statins and other lipid-lowering medications at baseline. Of the 4681 participants, 4592 had genotyping information, which was generated using a customized Affymetrix Axiom array including ~800,000 SNPs and further imputed to the 1000 Genomes reference panel (Phase 3) using IMPUTE v2.

## Measurement of lipid metabolites

A high-throughput targeted NMR metabolomics platform (*Soininen et al., 2015*) was used for quantification of circulating lipid metabolites in baseline plasma samples (Brainshake Laboratory at Kuopio, Finland). All metabolites were assayed simultaneously. Cases and controls were measured in random order, with laboratory staff blinded to case/control status. Of the 4681 participants, 137 had duplicated measurements. The median coefficient of variation for duplicates was 5.0% (interquartile range: 2.7–6.7%) (*Holmes et al., 2018*). Six traits covered by NMR spectroscopy were also measured using standard clinical chemistry assays including total cholesterol, LDL-C, high-density lipoprotein cholesterol (HDL-C), triglyceride (TG), apolipoprotein B, and apolipoprotein A1 (Wolfson Laboratory at University of Oxford, UK). There were high correlations between NMR and clinical chemistry measured traits, with the correlation ranging from 0.80 to 0.90 (*Holmes et al., 2018*).

## Definition and assessment of HLFs

We included five baseline lifestyle-related characteristics: smoking, alcohol consumption, dietary habit, physical activity, and body weight and fat to assess energy balance (*Lloyd-Jones et al., 2010*). In the baseline questionnaire, for smoking, we asked frequency, type, and amount of tobacco smoked per day for ever smokers, and years since quitting and reason to quit for former smokers. For alcohol consumption, we asked drinking frequency on a week, type of alcoholic beverage, and volume of alcohol consumed on a typical drinking day. For physical activity, we asked the usual type and duration of activities. The daily level of physical activity was calculated by multiplying the metabolic equivalent tasks (METs) value for a particular type of physical activity by hours spent on that activity per day and summing the MET hours for all activities. For dietary habit, we used a short qualitative food frequency questionnaire to assess habitual intakes of 12 conventional food groups (*Supplementary file 2G*). For adiposity level, trained staff measured weight, height, and WC with calibrated instruments. BMI was calculated as weight in kilograms divided by the square of the height in meters.

The HLFs that may be related to lower CHD risk were defined as follows: (1) never smoking; (2) moderate alcohol consumption: weekly but not daily drinking, or daily drinking less than 30 g of pure alcohol; (3) having $\geq 4$ of the total six healthy dietary habits that are particularly addressed in the Chinese dietary guidelines (2016) (*Yang et al., 2018*): consuming fresh vegetables every day, fresh fruits every day, red meat <7 days/week, soybean products $\geq 4$ days/week, fish $\geq 1$ day/week, and coarse grains $\geq 4$ days/week; (4) being physically active, i.e. having a sex-specific median or higher level of physical activity; (5) healthy adiposity levels: BMI in the range of 18.5–27.9 kg/m$^2$ (normal or overweight according to the standard classification specific for Chinese) and WC <90 cm in men and <85 cm in women (*Chen et al., 2018*; *Jia et al., 2010*).

## Genetic scores for *HMGCR* and *ACLY*

We constructed the genetic scores in the Chinese population with a previously adopted method (*Ference et al., 2019*). This approach has been used to accurately anticipate the results of several randomized trials that have evaluated lipid-lowering therapies (*Ference, 2018*). First, we tested the association of each variant within a 500 KB window on either side of the *HMGCR* gene in a linear regression model, with plasma LDL-C as dependent variable and age, sex, and the top 10 ancestry-informative principal components as covariates in 13,060 participants from the CKB cohort without overlapping with the lipidomic data set. All 13,060 participants were not using statin and other lipid-lowering medications at baseline. Second, we pruned the variant by keeping the top variants with most significant p-value and removed other variants that were correlated with the selected variant

($r^2 > 0.3$). Next, we tested the association between each remaining variant and LDL-C, conditional on previously selected variants and covariates, and selected the variant with the smallest p-value. We iteratively repeated this step until all variants were selected, removed due to linkage disequilibrium with a selected variant, or were not associated with LDL-C (p>0.05). The exposure allele for each selected variant was defined as the allele associated with lower plasma LDL-C. The weight for each variant was the conditional effect of that variant on LDL-C level in mmol/l adjusted for all other variants included in the score among the 13,060 participants.

We multiplied the number of exposure alleles that a participant inherited at each variant by their weights. We then summed these values to construct a weighted *HMGCR* genetic score for each participant in the present analysis. We used the same protocol to construct a weighted *ACLY* genetic score. We called this as genetic score based on Chinese population.

We also used previously constructed *HMGCR* and *ACLY* genetic scores (*Ference et al., 2019*) for comparison with studies in the European population. Three of the total nine variants included in the *ACLY* genetic score (rs113201466, rs145940140, and rs117981684) were monomorphism in the eastern Asian population. Only the other six variants were used to construct the genetic score for *ACLY*. Linear regression was used to estimate the effect of each variant on LDL-C level, with adjustment for age, sex, and the top 10 ancestry-informative principal components. We called this as genetic score based on European population. We also used the conditional effect reported from the previous study (*Ference et al., 2019*) to construct alternative weighted genetic scores for sensitivity analyses.

Variants included in the genetic scores and their association with LDL-C were provided in *Supplementary file 2H*.

## Statistical analysis

We classified participants into three groups according to the number of HLFs they adopted: zero to one, two to three, and four to five. All lipid metabolites were inverse normal transformed (SD = 1), which is useful for comparing variables expressed in different units. The associations between combined HLFs and lipid metabolites were assessed using linear regression adjusted for age (years), sex (male or female), fasting time (<8 or ≥8 hr), 10 study areas, education level (no formal or primary school, middle or high school, technical school or college or higher), and case/control status, with participants who adopted zero to one HLF as the reference group. Logistic regression was used to estimate odds ratios (ORs) of CHD per 1-SD higher lipid metabolite levels, adjusted for age, sex, fasting time, study areas, education level, and smoking status. The additional adjustment was also made for other HLFs (alcohol consumption, dietary habit, physical activity, and BMI) and prevalent diabetes, with results largely unchanged (data not shown).

We used the paramed package (*Emsley and Liu, 2013*) to perform causal mediation analysis (*Valeri and VanderWeele, 2013*) using parametric regression models. For each lipid metabolite, two models were estimated: (1) a model for the mediator (the lipid metabolite itself) conditional on exposure (the number of HLFs as a continuous variable) and covariates (age, sex, fasting time, study areas, and education level) in the control participants only (n = 1513); (2) a model for the risk of CHD conditional on exposure, the mediator, and covariates (age, sex, fasting time, study areas, and education level) in both CHD case and control participants (n = 2440). We also allowed for the presence of exposure-mediator interactions in the outcome regression model. We aimed to access how much of the total effect (TE) is due to neither mediation nor interaction; how much is due to interaction but not mediation; how much is due to both mediation and interaction; and how much of the effect is due to mediation but not interaction (natural indirect effect [NIE]). We used the delta method to estimate standard errors and confidence intervals. If exposure-mediator interactions did not exist, the proportion attributable to the NIE was calculated by dividing the NIE by TE on log odds scale, with 0 indicating no mediation effect. We also used the top five principal components of all lipid metabolites that explained ≥95% of the total variation to estimate the joint mediating effect of all lipid metabolites.

The *HMGCR* and *ACLY* genetic scores were also inverse normal transformed to facilitate comparisons. We estimated the association of a 2-SD increase in the *HMGCR* or *ACLY* score with lipid metabolites using linear regression, with adjustment for age, sex, study areas, and the top 10 genotype principal components. We further examined the joint association of HLFs and genetic scores with lipid metabolites, by classifying participants into four groups according to their genetic score (median cutoffs) and number of HLFs (zeo to two or three to five). We also examined whether the

association between HLFs and lipid metabolites differed by scores of *HMGCR*, *ACLY*, or their sum score, which were dichotomized according to the median cutoffs of the genetic scores. The tests for interaction were performed using likelihood ratio tests comparing models with and without the cross-product term.

All analyses were performed with Stata version 14.2 (StataCorp) and R software version 3.5.2 (R Foundation for Statistical Computing). p-values are presented as unadjusted for multiple testing unless otherwise indicated. For testing of multiple lipid metabolites, we used the FDR (*Benjamini and Hochberg, 1995*) <0.05.

## Acknowledgements

The most important acknowledgement is to the participants in the study and the members of the survey teams in each of the 10 regional centers, as well as to the project development and management teams based at Beijing, Oxford and the 10 regional centers.

## Additional information

### Funding

| Funder | Grant reference number | Author |
|---|---|---|
| National Key R&D Program of China | 2016YFC0900500 | Yu Guo |
| Wellcome Trust | 202922/Z/16/Z | Zhengming Chen |
| National Natural Science Foundation of China | 81390540 | Liming Li |
| National Natural Science Foundation of China | 81390544 | Jun Lv |

The funders had no role in study design, data collection and interpretation, or the decision to submit the work for publication.

### Author contributions

Jiahui Si, Formal analysis, Visualization, Writing - original draft, Writing - review and editing; Jiachen Li, Formal analysis, Validation, Writing - review and editing; Canqing Yu, Iona Millwood, Ling Yang, Robin Walters, Yiping Chen, Huaidong Du, Li Yin, Jianwei Chen, Data curation, Investigation, Writing - review and editing; Yu Guo, Data curation, Funding acquisition, Investigation, Project administration, Writing - review and editing; Zheng Bian, Junshi Chen, Data curation, Investigation, Project administration, Writing - review and editing; Zhengming Chen, Data curation, Supervision, Investigation, Project administration, Writing - review and editing; Liming Li, Conceptualization, Resources, Data curation, Supervision, Funding acquisition, Investigation, Project administration, Writing - review and editing; Liming Liang, Conceptualization, Supervision, Methodology, Writing - review and editing; Jun Lv, Conceptualization, Data curation, Supervision, Funding acquisition, Investigation, Methodology, Project administration, Writing - review and editing

### Author ORCIDs

Jun Lv (iD) https://orcid.org/0000-0001-7916-3870

### Ethics

Human subjects: The study protocol was approved by the Ethics Review Committee of the Chinese Center for Disease Control and Prevention (005/2004, Beijing, China) and the Oxford Tropical Research Ethics Committee, University of Oxford (025-04, UK). All participants provided written informed consent.

### Decision letter and Author response

Decision letter https://doi.org/10.7554/eLife.60999.sa1

Author response https://doi.org/10.7554/eLife.60999.sa2

## Additional files

### Supplementary files

- Source code 1. Association between lifestyle and metabolites.
- Source code 2. Association between metabolites and CHD.
- Source code 3. Joint effect of genetic score and HLFs on CHD.
- Source code 4. Mediation analysis.
- Source code 5. Construction of genetic scores.
- Source code 6. Simulation of mediation analysis.
- Supplementary file 1. Members of the China Kadoorie Biobank collaborative group.
- Supplementary file 2. Baseline characteristics according to case or control status, results of sensitivity analysis, and other useful materials in the methods section. (A) Age-, sex-, and study area-adjusted baseline characteristics of 4681 participants according to case or control status. (B) Sensitivity analysis of association between combined healthy lifestyle and lipid metabolites. (C) Associations of the *HMGCR* score with changes in the lipid metabolites, and subgroup analysis of the association between healthy lifestyle and lipid metabolites according to *HMGCR* score. (D) Associations of the *ACLY* score with changes in the lipid metabolites, and subgroup analysis of the association between healthy lifestyle and lipid metabolites according to *ACLY* score. (E) Associations of the sum of *HMGCR* and *ACLY* score with changes in the lipid metabolites, and subgroup analysis of the association between healthy lifestyle and lipid metabolites according to the sum score. (F) Associations of the genetic scores* with changes in the lipid metabolites. (G) Food frequency questionnaire used in the CKB study at baseline. (H) Variants included in the genetic scores and their associations with plasma low-density lipoprotein cholesterol.

- Transparent reporting form

### Data availability

According to the Regulation of the People's Republic of China on the Administration of Human Genetic Resources, we are not allowed to provide Chinese human clinical and genetic data abroad without an official approval. We are providing our syntax of statistical analysis and the output from the analysis.

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
