## [Decision Letter]

**Acceptance summary:**

The novel approach is a detailed lipidomic profile of 61 plasma metabolites by NMR at baseline in a large number of Chinese subjects associated with five individual lifestyle related factors ( dietary habit, exercise, non-smoking, adiposity and alcohol consumption) either singly or in combination and the impact of these on vascular disease after 20 years of follow up. Adherence as expected improved both the profile and outcomes.

**Decision letter after peer review:**

Thank you for submitting your article "Improved lipidomic profile mediates the effects of adherence to healthy lifestyles on coronary heart disease" for consideration by *eLife*. Your article has been reviewed by three peer reviewers, including Edward D Janus as the Reviewing Editor and Reviewer #1, and the evaluation has been overseen by Matthias Barton as the Senior Editor. The following individual involved in review of your submission has agreed to reveal their identity: Corey Giles (Reviewer #2).

The reviewers have discussed the reviews with one another and the Reviewing Editor has drafted this decision to help you prepare a revised submission.

Summary:

The authors report, in 4681 subjects, the detailed baseline lipidomic profile (61 plasma metabolites by NMR) associated with five individual healthy lifestyle related factors singly or in combination (dietary habit, exercise, non-smoking, adiposity and exercise) and relate these to the presence of coronary or cerebrovascular disease after approximately 20 years of follow up vs the baseline parameters for a further 1513 non affected matched control subjects. All measurements were without lipid lowering drugs.

They used regression to assess associations between health life-style factors (HLF); mediation analysis to assess mediating effects of metabolites on coronary heart disease HLF; pharmacomimetic gene scores to estimate the genetic effects of HMGCR and ACLY.

The beneficial effects of the five lifestyle characteristics are already well known so the main new data is the detailed lipidomic profile in a Chinese population and its implications. Participants who adhered to a combined healthy lifestyle were more likely to have a cardio-protective lipidomic profile which jointly mediated 14% of the protective effect of combined healthy lifestyle on CHD risk.

The manuscript is well written overall and has important clinical interpretation although there is an overload of information in the supplementary material. There are some issues of clarity in the text and some major limitations in the analysis which need to be addressed.

1).While the Title is clear and acceptable the statement in the Abstract – Healthy lifestyles included baseline smoking, alcohol consumption, dietary habit, physical activity, and adiposity levels – is not appropriate as some of these are unhealthy lifestyles or as in the case of adiposity are not even lifestyles. Better to use "Baseline lifestyle related characteristics" throughout the manuscript.

Also in the Abstract – significant mediation effects in the pathway from healthy lifestyles to CHD – is not correct as this says that healthy lifestyles cause CHD. If you delete "healthy" it’s then correct but even better is to leave your wording but change it to "CHD reduction".

2) In the last paragraph of the Introduction say something in one or two sentences about the cohort, CHD and the controls. Without this the Results section is difficult to understand because it precedes the Materials and methods section.

3) Results

At beginning please list the five Healthy lifestyle factors.

4) Mediation effects of lipid metabolites in the association between HLFs and CHD risk. In the first line you need to make it clear what is meant by CHD risk that is state it is the presence of coronary or cerebrovascular disease in members of the cohort during the follow up period. Also CHD means coronary heart disease so cerebro-vascular disease is not CHD. You need to use Coronary heart disease AND cerebrovascular disease throughout the paper in title, text, tables, figures. You need to also make it very clear that you have included intracerebral haemorrhage because this is not atherosclerotic in origin. Ideally you would show that excluding haemorrhagic stroke while retaining ischemic stroke does not substantially change your overall findings/implications.

5) Please provide the mean and SD number of years of follow-up for the cohort and specify how reliable and complete follow up was. This is important in your analysis. If incomplete then it’s a limitation to be discussed.

6) A major limitation of this study is the dichotomization of health factors, summing the number of factors for each person, then stratifying participants into one of three groups. I appreciate the authors intention to create an aggregate score for “health behavior”, but used in this context, it is indeed a limitation.

The HLF used in this study have all been linked with CHD. Dichotomizing them results in substantial loss of information. Summing across these scores then leads to a loss in specificity for the underlying biology. In effect, you may produce a variable that associates with CHD, but does not provide any more biological insight than what we knew over 50 years ago.

As shown in the supplement, much of the HLF could be driven by adiposity alone. This is after dichotomization, so I would expect even stronger results if the authors used raw BMI or WC. Some of the individual HLF have vastly different association strengths with metabolites, some even in the opposite direction to each other. The aggregate HLF just ends up losing power in this situation and hides the underlying biology. Associations of individual HLF (without dichotomization) provides information that can help improve our understanding of the metabolic consequences of these HLF. The current manuscript is lacking this insight.

Importantly, mediation analysis is no longer valid when the exposure variable is dichotomized. Causal mediation analysis will suggest a mediation effect even when this is not true. There is no getting around this, unfortunately. Whenever information is removed from the exposure, the putative mediating variable will attempt to explain it.

7) Alcohol consumption appeared to have a complex effect on the lipoprotein profile. From Table 1, around 28.3% of the study population with >= 4 healthy lifestyle factors were considered as moderate alcohol consumers. What was the general distribution of alcohol intake among study population, e.g., non-regular, previous regular, or heavy intake? This information should be given to better understand the effect of alcohol consumption.

8) The authors appear to have attempted a two-factorial analysis of gene scores by the HLF aggregate. It is now known that using the derived scores and an interaction term provides much more power for identifying independence or importance.

I am not sure why this analysis was performed: "We also examined whether the association between HLFs and lipid metabolites differed by scores of HMGCR, ACLY, or their sum score". Each genetic variant often has small effect sizes (much smaller than a typical dose of statin). Even the combined score would be very small compared to this. Assuming there isn't some underlying confounding, the inheritance of the genetic variants should be randomly distributed among the population. I do not believe there is any rationale that the association of metabolites with HLF should differ under these conditions.

9) The authors use the 1000 Genomes reference panel (Phase 3) to impute variants. Is this database (relative dearth of Chinese) appropriate to use for imputing variants in a Chinese population? The authors constructed genetic sores to mimic the effect of statins and ACLY inhibitors based on the Chinese population and also the European population. Use of the genetic scores based on the European population observed similar but weaker associations, especially for some lipid metabolites (small VLDL-TG and LDL-TG) (Supplementary file 1E).

---

## [Author Response]

Revisions for this paper:1) While the Title is clear and acceptable the statement in the Abstract – Healthy lifestyles included baseline smoking, alcohol consumption, dietary habit, physical activity, and adiposity levels – is not appropriate as some of these are unhealthy lifestyles or as in the case of adiposity are not even lifestyles. Better to use "Baseline lifestyle related characteristics" throughout the manuscript.Also in the Abstract – significant mediation effects in the pathway from healthy lifestyles to CHD – is not correct as this says that healthy lifestyles cause CHD. If you delete "healthy" it’s then correct but even better is to leave your wording but change it to "CHD reduction".

We thank the reviewer for pointing this out. We included adiposity measures as a lifestyle factor to assess energy balance, same as previous study (1). We have revised to use “baseline lifestyle related characteristics” or change the wording of each lifestyle throughout the manuscript to keep consistent. We also added “reduction” in the Abstract section.

2) In the last paragraph of the Introduction say something in one or two sentences about the cohort, CHD and the controls. Without this the Results section is difficult to understand because it precedes the Materials and methods section.

We thank the reviewer for pointing this out. We have added a brief description of the study population at the end of the Introduction section.

3) ResultsAt beginning please list the five Healthy lifestyle factors.

We have added to help better understanding.

4) Mediation effects of lipid metabolites in the association between HLFs and CHD risk. In the first line you need to make it clear what is meant by CHD risk that is state it is the presence of coronary or cerebrovascular disease in members of the cohort during the follow up period. Also CHD means coronary heart disease so cerebro-vascular disease is not CHD. You need to use Coronary heart disease AND cerebrovascular disease throughout the paper in title, text, tables, figures. You need to also make it very clear that you have included intracerebral haemorrhage because this is not atherosclerotic in origin. Ideally you would show that excluding haemorrhagic stroke while retaining ischemic stroke does not substantially change your overall findings/implications.

Of the total 4,681 participants, cases were those who had a newly developed fatal or nonfatal disease during follow-up: (1) CHD: fatal ischemic heart disease coded as ICD-10 I20-I25 and nonfatal myocardial infarction coded as I21-I23 (n=927); (2) ischaemic stroke: ICD-10 I63 or I69.3 (n=1,114); (3) intracerebral haemorrhage: ICD-10 I61 or I69.1 (n=1,127). Case status was defined as the disease first occurred in each participant. Common controls were selected by frequency matching to combined cases by age, sex, and study area (n=1,513).

In the analyses of HLFs and metabolites, we included all 4,681 participants and adjusted for case/control status to increase power. We also restricted analyses to control participants as a sensitivity analysis (Supplementary file 1B). In the analyses of metabolites and CHD, and corresponding mediation analyses, we excluded both ischaemic and haemorrhagic stroke. We apologize for the confusion because the Results section precedes the Materials and methods section. We have added a brief description of the study participants involved in this part of analyses in the “Result – Mediation effects” section and the last paragraph of the Introduction section.

5) Please provide the mean and SD number of years of follow-up for the cohort and specify how reliable and complete follow up was.This is important in your analysis. If incomplete then it’s a limitation to be discussed.

The mean follow-up duration of the cohort since baseline was 9.2 (1.4) years. We added the mean follow-up duration of the cohort in the Materials and methods – Study population section. The electronic linkage with the national health insurance (HI) claim databases started in 2011, which has become an important means of following up. Linkage to local HI databases has been achieved by 97% of the participants since 2014. By December 31 2015, of the all cohort participants, only 4875 (<1%) were lost to follow-up. The diagnosis adjudication has finished for 34,000 reported cases of ischemic heart disease by a review of hospital medical records. Overall, 88% of the diagnoses were confirmed. We have also added this information to the manuscript.

6) A major limitation of this study is the dichotomization of health factors, summing the number of factors for each person, then stratifying participants into one of three groups. I appreciate the authors intention to create an aggregate score for “health behavior”, but used in this context, it is indeed a limitation.The HLF used in this study have all been linked with CHD. Dichotomizing them results in substantial loss of information. Summing across these scores then leads to a loss in specificity for the underlying biology. In effect, you may produce a variable that associates with CHD, but does not provide any more biological insight than what we knew over 50 years ago.As shown in the supplement, much of the HLF could be driven by adiposity alone. This is after dichotomization, so I would expect even stronger results if the authors used raw BMI or WC. Some of the individual HLF have vastly different association strengths with metabolites, some even in the opposite direction to each other. The aggregate HLF just ends up losing power in this situation and hides the underlying biology. Associations of individual HLF (without dichotomization) provides information that can help improve our understanding of the metabolic consequences of these HLF. The current manuscript is lacking this insight.Importantly, mediation analysis is no longer valid when the exposure variable is dichotomized. Causal mediation analysis will suggest a mediation effect even when this is not true. There is no getting around this, unfortunately. Whenever information is removed from the exposure, the putative mediating variable will attempt to explain it.

Previous studies have examined the association between individual lifestyle related characteristics (without dichotomization) and lipidomic profile separately, including physical activity (2), alcohol consumption (3), BMI (4), and consumption of whole grain, fish and bilberries (5). In our study, we also performed analyses for individual factors (Figure 1—figure supplement 2-7). Our findings on the associations of lipid metabolites with individual lifestyle related characteristics like physical activity, adiposity, and alcohol consumption were generally consistent with previous studies. We agree with the reviewer that individual HLF may lead to different or even opposite effect on metabolites. Lifestyle related characteristics are also typically correlated with one another. Previous studies suggested that a large proportion of premature death (6) and increased risk of cardiometabolic diseases (7, 8) are attributed to the combination of these unhealthy lifestyles (dichotomized), providing important information on the maximum public health benefit that lifestyle intervention could achieve. A comprehensive analysis of the impact of adopting HLFs on the lipidomic profile is still lacking. Studies are warranted to examine the combined effect on metabolites as the mediators to cardiometabolic diseases. Our study has this unique opportunity to address this important question. Therefore, our main aim was to align with these previously used definitions and provide evidence of the combined effects of lifestyle related characteristics on lipidomic profile.

We agree with the reviewer that dichotomizing a continuous variable may lose information. A more sophisticated score with appropriated weight might show stronger association. However, a more straightforward definition would be easier to understand and adapted by the public. We have addressed this issue in the Discussion section.

Regarding the mediation analysis, we agree that when the exposure and mediator are highly correlated and the exposure is dichotomized with losing information, the mediator might show a biased association with the outcome even when it is not. However, this requires a particularly strong association between exposure and mediator. To demonstrate this phenomenon, we picked the strongest exposure-mediator association observed in our study (i.e., BMI-ApoB/ApoA1 with R^2^=0.137). We simulated the data with same sample size and case-control numbers, the same BMI-disease association (effect size OR=1.023) and the same BMI-metabolite association as BMI-ApoB/ApoA1 but no association between metabolite and disease outcome. We designed a series of simulations using different R^2^ (0.10–0.99), each repeating 100 times. The simulated data showed as R^2^ increases the mediator will have larger bias in association with the outcome (Author response image 1), and subsequently larger bias in the corresponding mediation effect (Author response image 2).

**Author response image 1. respfig1:** The p value frequency for the association between the mediator and the outcome among the 100 simulations.

**Author response image 2. respfig2:** The p value frequency for the mediation effect among the 100 simulations.

Author response table 1 shows the Bonferroni-corrected minimum p value among the 100 simulations. The BMI-metabolite association needs to be increased to R^2^=0.80 to generate a falsely significant mediation effect to achieve Bonferroni-corrected p value < 0.05. This is far from the realistic association between HLFs and metabolite in our study (the strongest R^2^=0.137 for BMI – ApoB/ApoA1). To make a direct comparison, we tested the mediation effect for dichotomized BMI – ApoB/ApoA1 – CHD and the observed mediation p value was less than 2E-16 in our study, far from that would have been biased due to loss of information. We have discussed this issue in the Discussion section.

**Author response table 1. resptable1:** The Bonferroni-corrected minimum p value in 100 simulations.

	Bonferroni-corrected minimum p value for the mediator – outcome association	Bonferroni-corrected minimum p value for the mediation effect
R^2^=0.10	0.215	0.227
R^2^=0.137	0.258	0.216
R^2^=0.15	0.124	0.117
R^2^=0.20	0.133	0.189
R^2^=0.30	0.114	0.154
R^2^=0.40	0.153	0.198
R^2^=0.50	0.290	0.234
R^2^=0.60	0.217	0.247
R^2^=0.70	0.057	0.096
R^2^=0.80	0.020	0.047
R^2^=0.90	0.014	0.018
R^2^=0.99	0.011	0.005

7) Alcohol consumption appeared to have a complex effect on the lipoprotein profile. From Table 1, around 28.3% of the study population with >= 4 healthy lifestyle factors were considered as moderate alcohol consumers. What was the general distribution of alcohol intake among study population, e.g., non-regular, previous regular, or heavy intake? This information should be given to better understand the effect of alcohol consumption.

In the present study, around 77.5% of the participants reported drinking less than weekly at baseline. Please find the detailed distribution of alcohol consumption in Author response table 2.

**Author response table 2. resptable2:** 

	Detailed distribution	No. of participants	Percent
Non-regular alcohol consumption	Not regular (less than weekly) drinker	3,626	77.5
	Former regular drinker	205	4.4
Moderate alcohol consumption	Weekly but not daily drinker	439	9.4
	Daily drinker with <15g pure alcohol per day	3	0.1
	Daily drinker with 15-29g pure alcohol per day	80	1.7
Heavy alcohol consumption	Daily drinker with 30-59g pure alcohol per day	120	2.6
	Daily drinker with ≥60g pure alcohol per day	208	4.4

8) The authors appear to have attempted a two-factorial analysis of gene scores by the HLF aggregate. It is now known that using the derived scores and an interaction term provides much more power for identifying independence or importance.I am not sure why this analysis was performed: "We also examined whether the association between HLFs and lipid metabolites differed by scores of HMGCR, ACLY, or their sum score". Each genetic variant often has small effect sizes (much smaller than a typical dose of statin). Even the combined score would be very small compared to this. Assuming there isn't some underlying confounding, the inheritance of the genetic variants should be randomly distributed among the population. I do not believe there is any rationale that the association of metabolites with HLF should differ under these conditions.

Because HLF and statin both have the potential impact on lowering cardiometabolic disease-associated lipids, we performed this analysis to explore the potential joint and interaction effect of HLFs with lipid-lowering drugs on lipid metabolites. Our results demonstrated the independent effect of both HLFs and lipid-lowering drugs on metabolites. Importantly we observed that when simultaneously adjusted in the model, they are targeting different components of the lipidomic profile. Similar to the intuition of the reviewer, we observed HLFs showed no interactions with neither genetic scores (to mimic the effect of HMGCR and ACLY) in its effect on lipid metabolites (Supplementary file 1C-1E). Our results suggest that the pathways affecting LDL and VLDL might be different. We prefer to keep these results to give a comprehensive understanding of the underlying pathway of HLFs and lipid-lowering drugs.

9) The authors use the 1000 Genomes reference panel (Phase 3) to impute variants. Is this database (relative dearth of Chinese) appropriate to use for imputing variants in a Chinese population? The authors constructed genetic sores to mimic the effect of statins and ACLY inhibitors based on the Chinese population and also the European population. Use of the genetic scores based on the European population observed similar but weaker associations, especially for some lipid metabolites (small VLDL-TG and LDL-TG) (Supplementary file 1E).

In our study, we used the 1000 Genomes Project reference panel (Phase 3, all populations) to impute ungenotyped SNPs in the CKB subjects (9, 10). The 1000 Genomes Project Consortium has demonstrated the imputation accuracy using all samples in 1000 Genomes Project as reference to impute other population data, the squared correlation between imputed and experimental genotypes was >95% for common variants in each population (including Chinese) (11). For both the HapMap and the 1000 Genomes Project, it has been a common recommendation that cosmopolitan reference panels be used to improve imputation accuracy for common and low-frequency variants (12, 13). We have provided details of imputation quality of the variants used to construct genetic scores (Supplementary file 1H). It is currently expected that genetic association effects learned from the European population sample are generally less applicable to other populations. Therefore we also developed the genetic score using effects learned from the CKB subjects.

References

1. Lloyd-Jones DM, Hong Y, Labarthe D, et al. Defining and setting national goals for cardiovascular health promotion and disease reduction: the American Heart Association’s strategic Impact Goal through 2020 and beyond. Circulation 2010;121:586–613.2. Kujala Urho M., Mäkinen Ville-Petteri, Heinonen Ilkka, et al. Long-term Leisure-time Physical Activity and Serum Metabolome. Circulation 2013;127:340–348.3. Würtz P, Cook S, Wang Q, et al. Metabolic profiling of alcohol consumption in 9778 young adults. Int. J. Epidemiol. 2016;45:1493–1506.4. Würtz P, Wang Q, Kangas AJ, et al. Metabolic Signatures of Adiposity in Young Adults: Mendelian Randomization Analysis and Effects of Weight Change. PLOS Med. 2014;11:e1001765.5. Lankinen M, Kolehmainen M, Jääskeläinen T, et al. Effects of Whole Grain, Fish and Bilberries on Serum Metabolic Profile and Lipid Transfer Protein Activities: A Randomized Trial (Sysdimet) Berthold HK, editor. PLoS ONE 2014;9:e90352.6. Li Y, Schoufour J, Wang DD, et al. Healthy lifestyle and life expectancy free of cancer, cardiovascular disease, and type 2 diabetes: prospective cohort study. BMJ 2020;368. Available at: https://www.bmj.com/content/368/bmj.l6669. Accessed February 18, 2020.7. Lv J, Yu C, Guo Y, et al. Adherence to a healthy lifestyle and the risk of type 2 diabetes in Chinese adults. Int. J. Epidemiol. 2017;46:1410–1420.8. Lv J, Yu C, Guo Y, et al. Adherence to Healthy Lifestyle and Cardiovascular Diseases in the Chinese Population. J. Am. Coll. Cardiol. 2017;69:1116–1125.9. Dai J, Lv J, Zhu M, et al. Identification of risk loci and a polygenic risk score for lung cancer: a large-scale prospective cohort study in Chinese populations. Lancet Respir. Med. 2019;7:881–891.10. Gan W, Bragg F, Walters RG, et al. Genetic Predisposition to Type 2 Diabetes and Risk of Subclinical Atherosclerosis and Cardiovascular Diseases Among 160,000 Chinese Adults. Diabetes 2019;68:2155–2164.11. Anon. A global reference for human genetic variation. Nature 2015;526:68–74.12. Howie B, Marchini J, Stephens M. Genotype Imputation with Thousands of Genomes. G3 Genes Genomes Genet. 2011;1:457–470.13. Li Y, Willer CJ, Ding J, Scheet P, Abecasis GR. MaCH: using sequence and genotype data to estimate haplotypes and unobserved genotypes. Genet. Epidemiol. 2010;34:816–834.